# Evaluating a large language model's ability to solve programming exercises from an introductory bioinformatics course

Stephen R. Piccolo[1]*, Paul Denny[2], Andrew Luxton-Reilly[2], Samuel H. Payne[1], Perry G. Ridge[1]

**1** Department of Biology, Brigham Young University, Provo, Utah, United States of America, **2** School of Computer Science, The University of Auckland, Auckland, New Zealand

* stephen_piccolo@byu.edu

**Data Availability Statement:** We created a GitHub repository (https://github.com/srp33/ChatGPT_Bioinformatics) that includes the data we collected in this study. The instructors' solutions have been

## Abstract

Computer programming is a fundamental tool for life scientists, allowing them to carry out essential research tasks. However, despite various educational efforts, learning to write code can be a challenging endeavor for students and researchers in life-sciences disciplines. Recent advances in artificial intelligence have made it possible to translate human-language prompts to functional code, raising questions about whether these technologies can aid (or replace) life scientists' efforts to write code. Using 184 programming exercises from an introductory-bioinformatics course, we evaluated the extent to which one such tool —OpenAI's ChatGPT—could successfully complete programming tasks. ChatGPT solved 139 (75.5%) of the exercises on its first attempt. For the remaining exercises, we provided natural-language feedback to the model, prompting it to try different approaches. Within 7 or fewer attempts, ChatGPT solved 179 (97.3%) of the exercises. These findings have implications for life-sciences education and research. Instructors may need to adapt their pedagogical approaches and assessment techniques to account for these new capabilities that are available to the general public. For some programming tasks, researchers may be able to work in collaboration with machine-learning models to produce functional code.

## Author summary

Life scientists frequently write computer code when doing research. Computer programming can aid researchers in performing tasks that are not supported by existing tools. Programming can also help researchers to implement analytical logic in a way that documents their steps and thus enables others to repeat those steps. Many educational resources are available to teach computer programming, but this skill remains challenging for many researchers and students to master. Artificial-intelligence tools like OpenAI's ChatGPT are able to interpret human-language requests to generate code. Accordingly, we evaluated the extent to which this technology might be used to perform programming tasks described by humans. To evaluate ChatGPT, we used requirements specified for 184 programming exercises taught in an introductory bioinformatics course at the

removed so that students cannot see the solutions for exercises that ChatGPT did not solve. Due to cell-size limitations in Microsoft Excel, we exported the spreadsheets to HTML. Researchers who wish to reuse the data can import the HTML files using R code and then can export it to other formats; an example is provided in our GitHub repository. The repository also includes a record of our conversations with ChatGPT. The files are in Markdown format. The researcher's portions of each conversation are prefixed with **Human:**. ChatGPT's portions of each conversation are prefixed with **Assistant:**. The code that we used to analyze the data and generate figures is available in our GitHub repository.

**Funding:** The authors received no specific funding for this work.

**Competing interests:** I have read the journal's policy and the authors of this manuscript have the following competing interests: After completing the analyses for this paper, OpenAI contacted us and offered credits to use their application programming interface for additional research. We accepted those credits but did not use the credits in this study, nor did this arrangement influence our conclusions in this study.

undergraduate level. Within 7 or fewer attempts, ChatGPT solved 179 (97.3%) of the exercises. These findings suggest that some educators may need to reconsider how they evaluate students' programming abilities, and researchers might be able to collaborate with such tools in research settings.

## Introduction

For decades, the life-sciences community has called for researchers to gain a greater awareness of "computing in all its forms" [1]. This need is now greater than ever. A 2016 survey of principal investigators from diverse biology disciplines revealed that almost 90% of researchers had a current or impending need to use computational methods [2]. Computers can help researchers formalize scientific processes [3], accelerate research progress [4], improve job prospects, and even learn biological concepts [5]. These opportunities have motivated the creation of interdisciplinary training programs, courses, workshops, and tutorials to teach computing skills in a life-sciences context [2,6–12]. In some circumstances, it is sufficient for researchers to understand computing concepts and learn to use existing tools; in others, learning to write computer code is invaluable [13]. A 2011 survey of scientists from many disciplines (other than computer science) found that researchers spent 35% of their time, on average, writing code [14]. Computer programming makes it possible to complete tasks not supported by existing tools, interface with software libraries, adapt algorithms based on custom needs, tidy data, and more [15–17]. In these applied scenarios, computer programs are often small [13] and used for only one project.

Scripting languages are well suited to such tasks because researchers can focus on high-level needs and worry less about memory management, code efficiency, and other technical details [15]. Python, a scripting language, has gained much acceptance among scientists [14] and programming educators [18], perhaps due to its relatively simple syntax [19] and the availability of libraries supporting common tasks [20–24]. However, learning to program is a daunting challenge for many researchers. Decades of research have sought to characterize common errors and identify effective ways for novices to learn programming skills [25–30]; much remains to be discovered.

Recent advances in artificial intelligence have shown promise for converting natural-language descriptions of programming tasks to functional code [31,32]. The first such *large language models* (LLMs) fine-tuned to generate code that captured widespread interest were OpenAI's Codex and DeepMind's AlphaCode [33,34]. These models were trained on millions of code examples, representing diverse programming tasks. In November 2022, OpenAI released *ChatGPT*, which uses an LLM fine-tuned with human feedback to generate natural dialogue-based text and code [35]. Researchers have speculated whether such models may be able to aid researchers—or even replace their efforts on basic programming tasks. For more complicated projects, LLMs might be able to assist in writing or debugging portions of the code. If successful in these settings, LLMs could reduce life scientists' time spent on programming, leaving more time for other research tasks.

We undertook a study to assess the extent to which an LLM can solve basic computer-programming tasks. By understanding LLMs' current capabilities and limits [36], we sought to gain perspective on their potential usefulness in life-sciences education and research. We used ChatGPT because it 1) was released recently, 2) is accessible with a Web browser, 3) can interact with users in a conversational manner, and 4) has garnered considerable attention among

academics, industry competitors, and the general public [37–43]. By January 2023, ChatGPT had over 100 million active users [44].

We evaluated and documented ChatGPT's effectiveness on Python-programming exercises from an introductory-bioinformatics course taught primarily to undergraduates. We evaluated how well ChatGPT could interpret the prompts and respond to human feedback to generate functional Python code. Here we describe quantitative and qualitative aspects of ChatGPT's performance, describe ways that ChatGPT could aid life scientists in research, and discuss implications for teaching and assessing students' programming capabilities in an educational context.

## Methods

### Programming exercises

Since 2012, Brigham Young University has offered an introductory bioinformatics course, *Introduction to Bioinformatics*. The course was designed for novice programmers who have an interest in biology. One learning outcome for the course is that "students will be able to create computer scripts in the Python programming language to manipulate biological data stored in diverse file formats." To facilitate skill development, the instructors created Python programming exercises, which serve as formative assessments. We used online datasets, articles, and tools to create the exercises [45–54]. Six exercises in the second assignment were derived from an online course [55]. To our knowledge, none of the other exercises were in the public domain at the time of our experiment; thus, it is unlikely that they were used to train the LLM before our testing. We used the "Jan 30 Version Free Research Preview" version of ChatGPT, which used version 3.5 of the Generative Pre-trained Transformer (GPT) model.

The exercises are organized into 19 assignments, each designed to teach a particular concept. Students complete the assignments and exercises in a defined sequence. The assignments cover 1) relatively simple tasks like declaring and using variables, performing mathematical calculations, and writing conditional statements; 2) medium-difficulty tasks like working with strings, lists, loops, dictionaries, and files; and 3) more advanced tasks like writing regular expressions, manipulating tabular data, and creating data visualizations. Other assignments give students practice with techniques that they learned in previous assignments. At specific times throughout the course, students complete additional programming exercises as summative assessments (exams), culminating in an end-of-course summative assessment. We excluded the summative assessments from this study.

The programming exercises are delivered via CodeBuddy, a Web-based application that acts as an automated grader (https://github.com/srp33/CodeBuddy). For each exercise, students receive a prompt describing the problem's context and requirements. The prompt sometimes includes basic code that students can use as a starting point. Each exercise has at least one test, including inputs and expected outputs. When applicable, the inputs consist of data file(s) provided within the prompt. The expected outputs may be text based (n = 179) or image based (n = 5). To generate the expected outputs, the instructor provides a solution; CodeBuddy executes the code and stores the output. The output of the student's code must match the expected output exactly. Students can make multiple attempts, as needed, without penalty. In many cases, instructors provide test(s) for which the inputs and/or expected outputs are hidden; this helps to prevent students from writing code that does not address the stated requirements. We excluded these tests from the study to maintain consistency with what students see; we manually verified whether ChatGPT-generated code met the requirements.

We used *openpyxl* (version 3.1.0) [56] to create a spreadsheet with information about each exercise. One column contains the prompt for each exercise, including the instructions and a

```
Here is a new Python programming task.

Please write a Python function called "addNumbers" that calculates the sum of two
numeric values. The function should accept two arguments (the two numbers to be
summed) and return the sum of these values.

Below is a description of a test to verify your code.

For this test, the following code will be executed after your code:

    print(addNumbers(2, 2))

Below is the expected output:

    4

Below is a description of a test to verify your code.

For this test, the following code will be executed after your code:

    print(addNumbers(0, 0))

Below is the expected output:

    0

Below is a description of a test to verify your code.

For this test, the following code will be executed after your code:

    print(addNumbers(-1, 5001))

Below is the expected output:

    5000

Below is a description of a test to verify your code.

For this test, the following code will be executed after your code:

    print(addNumbers(-5000.0, 5000.0))

Below is the expected output:

    0.0

Below is a description of a test to verify your code.

For this test, the following code will be executed after your code:

    print(addNumbers(5.5, 1))

Below is the expected output:

    6.5
```

**Fig 1. Example prompt for a programming exercise delivered to ChatGPT.**

summary of each test. Fig 1 shows an example of how the prompts were structured. For image-based tests, we did not include the expected outputs because ChatGPT does not accept images as input. To make the prompts more understandable to ChatGPT—and to mimic what students or researchers might do—we added natural-language transitions between each section of the prompt. Other columns in the spreadsheet include the instructors' solutions and flags indicating whether each exercise was biology oriented. Many exercise prompts provide biology-based scenarios such that a basic understanding of biology concepts is helpful when interpreting the prompts.

### Evaluation approach

We initiated a conversation with ChatGPT for each assignment. For one exercise at a time, we copied the prompt into ChatGPT's Web-based interface. To assess functional correctness, we copied ChatGPT's generated code into CodeBuddy. If the code did not pass all the tests, we continued the conversation with ChatGPT. In these interactions, we took the stance of a naive programmer who wishes to obtain functional code but is not necessarily able to provide detailed feedback about the code itself. We allowed ChatGPT a maximum of ten attempts per exercise. As we interacted with ChatGPT, we used the spreadsheet to record the dates of the interactions, ChatGPT's generated code (its final attempt), the number of passed tests, the number of attempts made by ChatGPT, and comments describing our interactions. When our interactions with ChatGPT suggested that a prompt lacked clarity, we slightly modified the prompt and updated the spreadsheet accordingly.

After completing our evaluation of ChatGPT, Google released *Bard*, a Web-based application that uses an LLM to generate text and code. We tested 66 of the 184 exercises using Bard (version: 2023.06.07). These exercises consisted of the first and last exercises in each assignment and all exercises in the following assignments: "01—Declaring and Converting Variables," "07—Problem Solving," "13—Advanced Functions & Additional Practice," and "19—Additional Practice."

When executing the Python code, we used version 3.8 of Python. To generate the manuscript figures and perform statistical analyses, we used the R statistical software (version 4.0.2) and the *tidyverse* packages (version 1.3.2) [57,58]. All statistical tests were two sided.

## Results

After our filtering steps, 184 Python programming exercises were available for testing. ChatGPT successfully solved 139 (75.5%) of the exercises on its first attempt. When it was unsuccessful on the first attempt, we engaged in a dialog with ChatGPT, allowing up to 10 interactions. Table 1 summarizes these interactions. In 26 instances, we indicated that the code had resulted in a runtime error, and we provided the error message to ChatGPT. More commonly, the generated code's output did *not* match the expected output, either due to a logic error (n = 44) or a simple formatting issue (n = 17). In many cases—typically after three or more interactions for a given exercise—we restated the original prompt (n = 30), a modified version of the prompt (n = 11), or simply asked the model to try again. Rarely (n = 7), we provided a suggestion about the code itself (e.g., to change a function name or to use a different parameter). Still, we never provided code with our feedback.

Of the 45 exercises that did not pass on the first attempt, ChatGPT solved 27 within 1 or 2 subsequent attempts. Within 7 or fewer attempts total, ChatGPT solved 179 (97.3%) of the

**Table 1. Summary of interactions between the human user and ChatGPT.** For the exercises that ChatGPT failed to solve on the first attempt, we categorized the subsequent interactions between the human user and the model. This table indicates the frequency of each interaction type.

| Interaction type | Count |
|---|---|
| Indicated logic error | 44 |
| Restated original prompt | 30 |
| Described runtime error | 26 |
| Described simple formatting issue | 21 |
| Requested simply to try again | 12 |
| Provided modified prompt | 11 |
| Suggested simple code tweak | 7 |

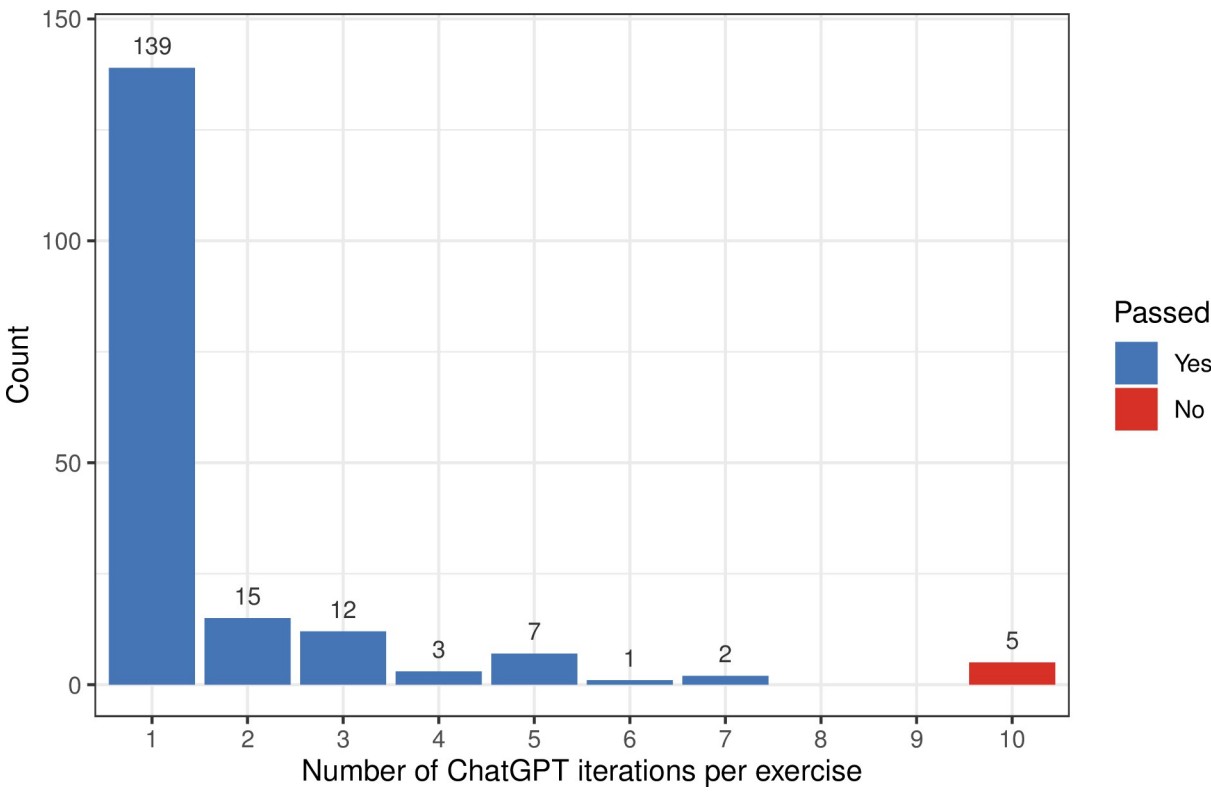

**Fig 2. Number of ChatGPT iterations per exercise.** For each exercise prompt, we gave ChatGPT up to 10 attempts at generating a code solution that successfully passed the tests. The counts above each bar represent the number of exercises that required a particular number of attempts.

exercises (Fig 2). As summarized in Table 2, the five unsolved exercises are delivered in the middle or end of the course; each requires students to combine multiple types of programming skills. One of these is the course's final exercise; the instructors' solution uses 61 lines of code, nearly twice as many as any other solution. For the remaining exercises that failed, ChatGPT came close to passing the tests. Its solutions resulted in logic errors or runtime errors or produced outputs that did not match the expected outputs exactly.

**Table 2. Summary of exercises that ChatGPT did not solve.** ChatGPT failed to solve 5 of the exercises within 10 attempts. This table summarizes characteristics of these exercises and provides a brief summary of complications that ChatGPT faced when attempting to solve them.

| Assignment | Exercise | Prompt summary | Skills emphasized | Failure summary |
|---|---|---|---|---|
| 09—Strings | 05—Is CpG island | Count the proportion of a DNA sequence that consists of 'CG' nucleotide pairs | Manipulating strings; performing mathematical calculations | Failed on one test representing an edge case (exactly 10% of the sequence were CG pairs) |
| 14—Regular Expressions 1 | 06—Find words that start with a vowel | Find words in a biological text that begin with vowels | Writing regular expressions; reading files | Trouble dealing with extra spaces or punctuation marks |
| 15—Regular Expressions 2 | 06—Switch column order | Switch the first two columns in a tab-delimeted text file | Writing regular expressions or using lists; reading files; writing files | Runtime errors, logic errors dealing with + or—portion of blood types |
| 19—Additional Practice | 08—Make inducible promoter—Part B | Identify in-frame start and stop codons in an mRNA sequence | Manipulating strings; reading files; using complex iteration logic | Failed to follow the instructions to look for the start codon in frame |
| 19—Additional Practice | 10—Make inducible promoter—Part D | Identify restriction-enzyme binding sites upstream of a gene in a DNA sequence | Reading files; writing regular expressions; using complex iteration logic; using lists; using dictionaries; using conditionals | Runtime errors, various logic errors, failure to fully comprehend the prompt |

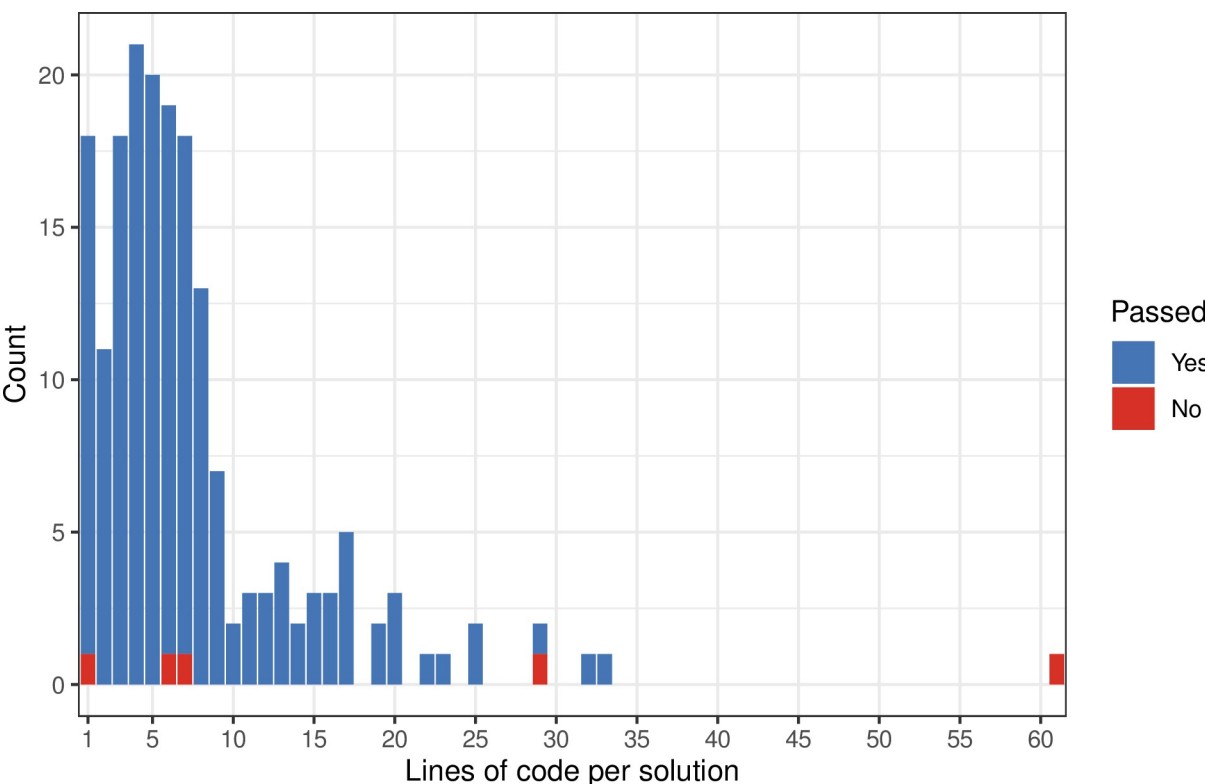

**Fig 3. Lines of Python code per instructor solution.** Course instructors provided a solution for each exercise. This plot illustrates the number of lines of code for each solution, after removing comment lines.

We used statistics to understand more about the scenarios in which ChatGPT either succeeded or failed. First, we used the length of the instructors' solutions as an indicator for difficulty level. After removing comments (inline descriptions of how the code works) and blank lines, we compared the number of lines of code between the exercises that ChatGPT solved and those that it did not (Fig 3). The median for passing solutions was 6, and the median for non-passing solutions was 7; this difference was not statistically significant (Mann-Whitney U p-value: 0.2836). The lengths of the instructors' solutions were significantly (positively) correlated with the lengths of ChatGPT's solutions (Fig 4), both for the number of characters (Spearman's rho = 0.89; p-value < 0.001) and the number of lines (Spearman's rho = 0.83; p-value < 0.001). Another indicator of difficulty level is the exercise-prompt length. For passing solutions, the median was 2019 characters, while the median was 9115 for non-passing solutions. Although this difference was not statistically significant (Mann-Whitney U p-value: 0.1021), it is consistent with a recent study of computer science exercises [32].

The number of attempts provides additional insight into ChatGPT's capabilities but should be cautiously interpreted because ChatGPT exhibits stochasticity. Whether ChatGPT provides a correct answer on the first or a later attempt, eventual success shows that its probabilistic model can aid users. However, a smaller number of attempts might suggest an ability to formulate a valid response more readily, thus requiring less time by the user. The number of attempts per exercise was significantly correlated with the length of the instructors' solution (rho = 0.234; p = 2.2e-16) and the length of the prompt (rho = 0.31; p = 2.2e-16). These correlations held, whether or not we considered the five exercises that ChatGPT failed to solve.

Of the 184 prompts, 98 (53.3%) were framed in a biological context. Of the five exercises that ChatGPT did not solve, four were framed in a biological context (Fisher's exact test p-

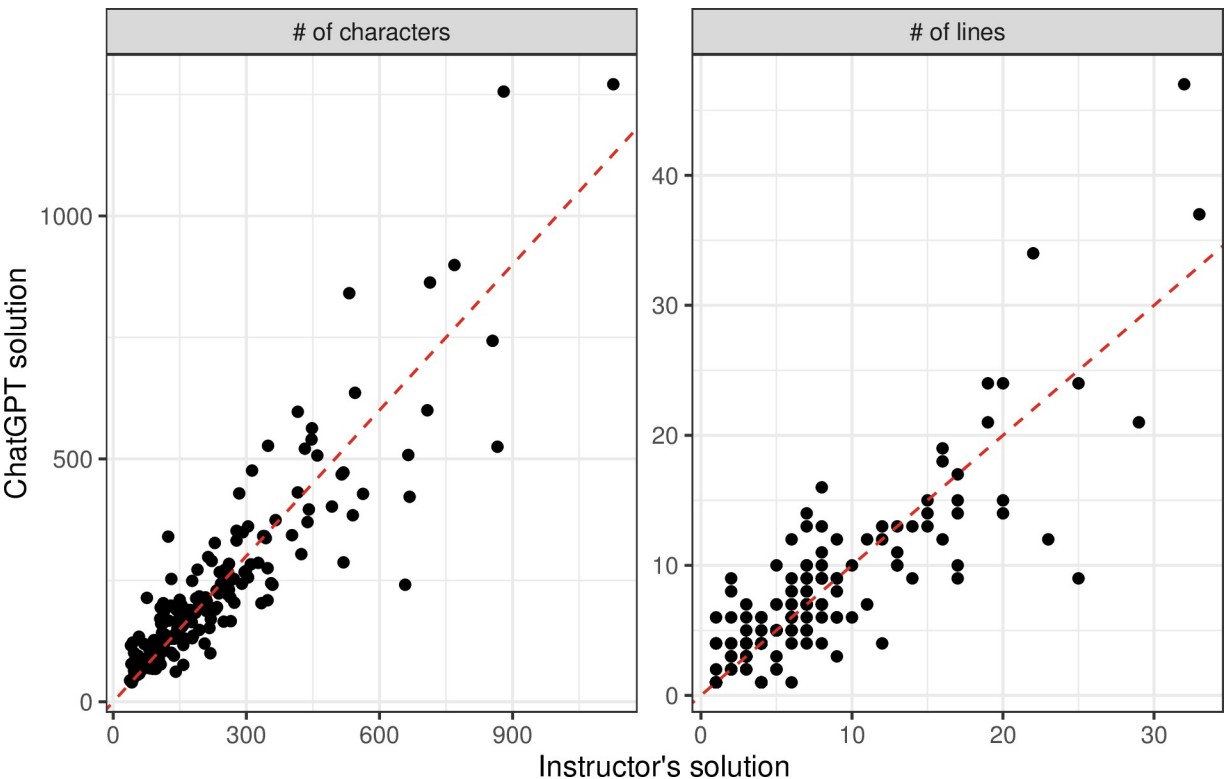

**Fig 4. Comparison of code-solution lengths for instructor solutions versus ChatGPT solutions.** This illustrates the relationship between A) the number of characters or B) the number of lines of code, for each exercise, after removing comment lines. The dashed, red line is the identity line.

value = 0.37). The median length (characters) of biology-oriented prompts was 3203, whereas the median was 1437 for the remaining prompts (Mann-Whitney U p-value = 1.6e-09). In the course, we frequently use biological data (e.g., genome sequences, medical observations, narrative text) to teach analysis skills and make the exercises more authentic. We included these data so that ChatGPT could evaluate the files' structure. For 24 exercises, the prompt size exceeded the maximum allowed by ChatGPT. After we truncated the data to the first few lines, ChatGPT was successful at solving all of these exercises. On four other occasions, we shortened parts of the prompt as we interacted with ChatGPT to attempt to provide clarity. For example, we shortened the descriptions of how the code would be tested. ChatGPT eventually solved two of these four exercises.

We note additional challenges that ChatGPT faced when interpreting the programming prompts. On 17 exercises, ChatGPT used correct logic but produced outputs that were different from the expected outputs (for example, "Number of worms in the last box: 5" instead of "5"). Eventually, ChatGPT solved all of these exercises. On 25 exercises, ChatGPT generated code that produced logic errors; it eventually solved 20 of these exercises. On 10 exercises, ChatGPT generated code that produced runtime errors (exceptions); it eventually solved 8 of these exercises. On two exercises, ChatGPT generated passing code that did not directly address the prompt. For example, in one case, the prompt called for using a regular expression (text-based pattern matching), but ChatGPT used iteration logic instead; we marked these exercises as passing because the automatic grader did not verify which type of logic they used. On five occasions, we noted parts of the prompt that may have been ambiguous. We clarified these prompts; subsequently, ChatGPT solved four of these exercises.

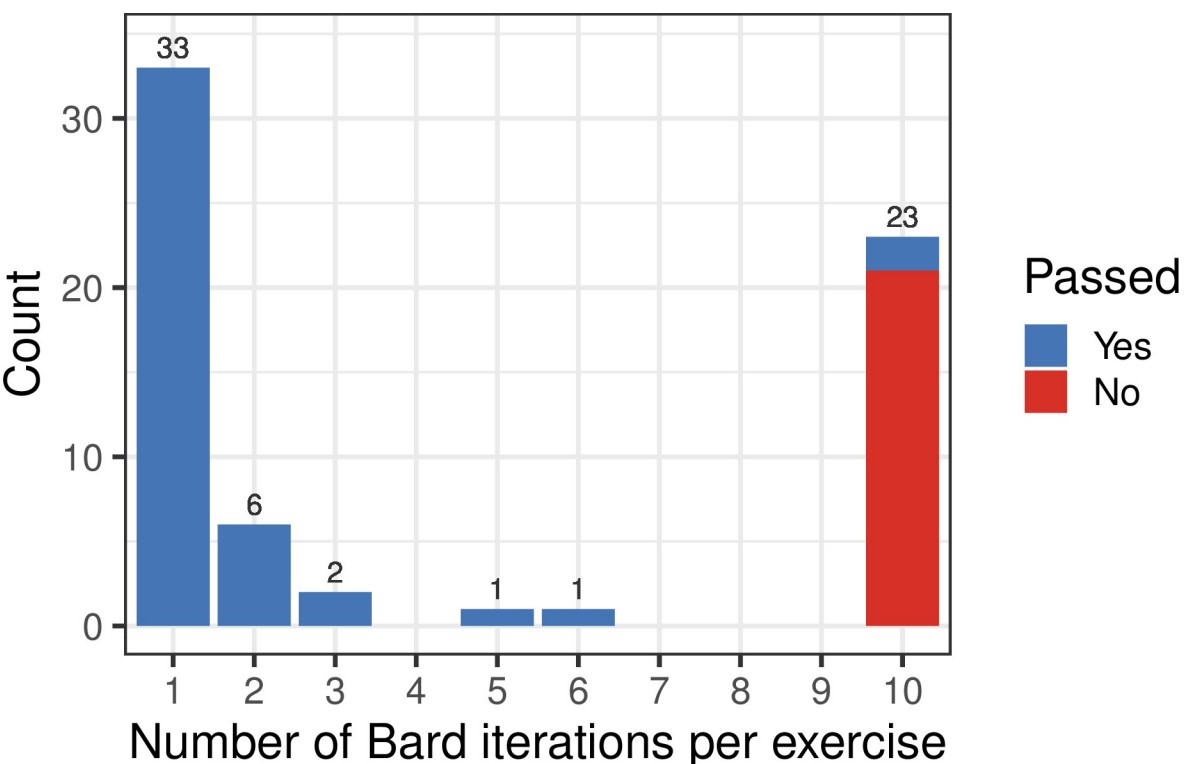

**Fig 5. Number of Bard iterations per exercise.** For 66 exercise prompts, we gave Google Bard up to 10 attempts at generating a code solution that successfully passed the tests. The counts above each bar represent the number of exercises that required a particular number of attempts.

In using ChatGPT to solve these programming problems, we observed several practical issues that may impact the value offered by ChatGPT to researchers and students. For 13 exercises that did not pass on the first attempt, we asked ChatGPT to try an alternative approach, and/or we re-delivered the original prompt. In these cases, we sought to take advantage of its stochastic nature, perhaps resulting in code that used a considerably different strategy. ChatGPT eventually passed 11 of these 13 exercises. For 41 (22.2%) exercises in total, ChatGPT used at least one programming technique that would have been unfamiliar to most students in the course. Many of these techniques are never taught in the course, whereas others are introduced in later units. Finally, following an unsuccessful first attempt for six exercises, ChatGPT generated code that did *not* address the original prompt. Conceivably, the model "forgot" earlier parts of the conversation or was "distracted" by subsequent inputs. Eventually, it solved all of these exercises.

Although the focus of this study was not to compare LLMs, we wished to approximate how well our findings would generalize to another LLM. We evaluated Google Bard's code-generation ability for 66 of the Python exercises. Bard solved 33 (50.0%) within one attempt and 45 (68.2%) within 10 attempts (Fig 5).

## Discussion

These findings demonstrate that modern LLMs can solve many basic Python programming tasks, often in a biological context. On educational assessments requiring basic programming skills, students might seek help from LLMs when it is available. Additionally, in some settings, researchers might be able to rely on LLMs' abilities to translate natural-language descriptions

to code. Researchers have already begun to explore this capability in practice [59]. We anticipate that as the models evolve, students and researchers will increasingly author programming prompts in addition to code.

Anecdotally, we have found that authoring programming prompts is not always easy. During our evaluations, communicating with the model was cognitively taxing at times. In addition, these conversations were sometimes awkward. Although ChatGPT can retain a memory of previous interactions, its default response was to provide a solution; in many cases, it might have been more helpful for ChatGPT to request additional information or clarification regarding the problem. It was often more effective for us to restate the original prompt than engage in a back-and-forth dialog. On a positive note, ChatGPT was exceptionally effective at determining which parts of a given prompt were most informative; for example, it seemed to identify relevant aspects of biological context and ignore extraneous details. Currently, LLMs do not execute code; thus, they often cannot predict the output of code [60]. This is one area in which human feedback remains critical.

For 60+ years, researchers have been working to automate program synthesis [61,62]. Recent efforts have focused on training neural networks on large code repositories [33,34,63,64]. Our results show that ChatGPT represents a considerable advance compared to prior models. Chen, et al. [33] evaluated Codex's ability to solve short- to medium-length programming exercises (median solution = 5.5 lines of code). When delivering the prompts, they used "docstrings" (structured descriptions of functions). Codex was successful for 28.8% of these exercises in a single attempt; when making 100 attempts per exercise, it solved 77.5% of the exercises [33]. In an additional study, Austin, et al. used a different set of exercises that were either mathematical or focused on core programming skills [60]. In contrast to Chen, et al., they used natural-language prompts (one or a few sentences). The solutions had a median length of 5 lines. Using various LLMs, they solved as many as 83.8% of the mathematical problems and 60% of the remaining problems (within 100 attempts). For a subset of the problems, they provided human-language feedback to the models (up to four interactions); maximum accuracy was 65%. In an educational context, Finnie-Ansley, et al. showed that Codex could solve 82.6% of programming exercises from an introductory computer-science course within 10 attempts and that the model would have ranked among the top quartile of students in the course [31]. Finally, in a similar approach to ours, Denny, et al. used Copilot (a development environment plug-in powered by OpenAI's Codex model) to solve 166 programming exercises designed for novice computer science students [65]. For problems that initially failed, they observed similar improvements in the model's performance through natural-language modifications to the prompts. However, only 80% of the problems were ultimately solved. Given that the problems they analyzed were also designed for novices, the superior performance we observed may be due to improvements in the models themselves in the intervening six months.

Aside from the use of ChatGPT, our work differs from prior evaluations in scope and context. Previous studies used exercises that evaluated the models' abilities to solve mathematical problems or to use core programming skills like processing lists, processing strings, or evaluating integer sequences. Our exercises required similar techniques and higher-level tasks like parsing data files, writing data files, creating graphics, and using external Python packages. Furthermore, more than half of our exercises were framed in a biological context. LLMs may be most helpful for routine tasks that appear frequently in training sets and only need to be modified for a particular purpose; however, our results show that LLMs can be used in new and diverse contexts as well.

Our findings have important implications for education. Unless LLMs demonstrate an ability to replace all human programming efforts, it will remain necessary for students (and

others) to gain programming skills [66]. In our course, preventing students from using LLMs on formative assessments (homework) would be impossible. However, summative assessments (exams) are the primary way we determine students' final grades; these assessments are invigilated, and students cannot access the Internet. Therefore, we retain confidence in the validity of grades determined under secure assessment conditions. Extensive practice is a critical part of learning to write code [67,68]. Thus, if students rely on LLMs to generate answers to formative assessments without first devising their own solutions, they may be more likely to perform poorly on summative assessments. Indeed, over-reliance by novices was a key risk identified by Chen et al. when releasing the Codex model [33]. With the ease of use and wide availability of tools like Copilot and ChatGPT, novices may quickly learn to rely on auto-suggested solutions without thinking about the computational steps involved—or reading problem statements carefully. Furthermore, if students copy and paste code without understanding it—as has been observed for an online forum [69]—they may underperform on summative assessments. One way that instructors could counter this behavior is to generate student-specific questions about their code. Lehtinen, et al. used an LLM to generate multiple-choice questions about code that students had submitted in an introductory-programming course [70]. Students who struggled to answer these questions were more likely to perform poorly or drop the course. LLMs also provide opportunities to make learning processes more efficient. For example, to aid students on programming exercises, the instructor could allow access to an LLM, which could act as an intelligent tutor. When a student struggled to complete a given exercise, the tutor could ingest the student's code and the exercise requirements and offer suggestions [71–73]. Doing so may reduce the need for instructors or teaching assistants to provide help. Finally, instructors might be able to use LLMs when creating new exercises to evaluate whether their prompts are clear.

It remains essential to have a human in the loop to evaluate the outputs of LLMs. Students and researchers must be competent at code comprehension and code evaluation. LLMs often produce code that does not meet the stated requirements; additionally, edge cases may not be specified as part of prompts. As a (simple) validation, we compared ChatGPT's output against the expected outputs that we had defined before beginning our study. This approach aligns with the educational context that we considered (an introductory course). However, in subsequent courses and the "real world," other types of validation would be necessary. Educators may need to shift pedagogical practice toward ensuring that students can understand code that has been generated, evaluating whether generated code meets specifications, debugging generated code, adapting code to different library versions, etc. Fig 6 provides recommendations on how to use LLMs effectively in an educational or research context.

We deliberately chose to allow ChatGPT up to 10 attempts to solve each exercise. Firstly, this criterion aligns with our pedagogical approach. The exercises we tested are formative. Accordingly, failing, receiving feedback, and re-attempting are part of the learning process [67,68,74]. Secondly, allowing multiple attempts reflects how biologists could use LLMs in research. If LLM-generated code does not function correctly on the first attempt, the researcher could ask the model to revise or generate a new solution. Thirdly, allowing multiple attempts per exercise is consistent with what others have reported [31,60].

Our study has several limitations. We applied one particular version of one LLM to all 184 exercises. We applied a second LLM (Google Bard) to a subset of the exercises. We do not know how our findings would generalize to other models or versions; however, the performance of LLM-based code generators will likely continue to improve as model sizes increase. The programming exercises we evaluated do not necessarily represent skills that would be taught in other introductory bioinformatics courses or used broadly in bioinformatics research. We used the Python programming language; our findings might not generalize to

**Do:**

- **Clearly state requirements**. Be specific about the task you want the LLM to perform. Explaining a bit about the big-picture goal may lead to more relevant code suggestions.

- **Start simple**. Develop your process by first using the model to generate relatively small and straightforward code snippets.

- **Review outputs**. LLMs can make mistakes or produce suboptimal code. Ensure that the generated code meets your requirements and follows sound coding practices.

- **Understand limitations**. LLMs may poorly understand your prompts, generate solutions with security vulnerabilities, or produce code that executes slowly. Use LLMs as assistants, not as substitutes for your own judgment.

**Don't:**

- **Overcomplicate prompts**. Avoid overly complex or ambiguous prompts. Simple and clear instructions may be more likely to produce effective code.

- **Rely blindly on outputs**. Always validate the code to make sure it meets your requirements. Identify potential problems, including security issues.

- **Use code that you do not understand**. Even if generated code meets your functional requirements, it might use a technique that you do not understand. Use it as a learning experience or request a simpler solution.

- **Share sensitive data**. Never include passwords, personal information, or proprietary secrets in prompts. Language models are not guaranteed to maintain data privacy.

**Fig 6. Recommendations for using LLMs to generate code.**

other languages. Future studies can shed additional light on how LLMs might be helpful for bioinformatics education and research.

Another limitation is that our evaluation process was subjective. When the initially generated solutions did not pass, the human user judged which types of feedback would be most helpful in each interaction. Other users would have interacted differently with the models. Furthermore, the human user was not a student but an instructor with 25 years of programming

experience and 15 years of Python experience. In our attempt to mimic novice programmers, we rarely suggested tweaks to the code (Table 1); perhaps students would have described problematic aspects of generated code more (or less) frequently. Additionally, students might have provided more (or less) context to the models about runtime errors that occurred.

In this study, we provide evidence that dialog-based LLMs, such as ChatGPT, can aid in solving basic programming exercises, with or without biological relevance. However, despite generally excellent performance, much remains to be learned about how these models can replace human programming efforts. In an authentic research setting, where an auto-grader cannot provide instant feedback on the correctness of model-generated code, there is a risk that relying on their outputs may produce erroneous results. Nevertheless, our findings have important implications for educators and researchers who seek to incorporate programming skills into their work. With the help of machine-learning models, instructors may be able to provide more personalized and efficient feedback to students, and researchers might be able to accelerate their work.

## Acknowledgments

Brandon Pickett, Justin Miller, Corinne Sexton, Ashlyn Powell, and Eric Upton-Rowley contributed to programming exercises that were used in this study.

## Author Contributions

**Conceptualization:** Stephen R. Piccolo, Paul Denny, Andrew Luxton-Reilly.

**Formal analysis:** Stephen R. Piccolo.

**Investigation:** Stephen R. Piccolo.

**Resources:** Stephen R. Piccolo, Samuel H. Payne, Perry G. Ridge.

**Software:** Stephen R. Piccolo.

**Visualization:** Stephen R. Piccolo.

**Writing – original draft:** Stephen R. Piccolo, Paul Denny, Andrew Luxton-Reilly, Samuel H. Payne.

**Writing – review & editing:** Stephen R. Piccolo, Samuel H. Payne, Perry G. Ridge.

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
