## [Decision Letter · Decision Letter 0]

16 Jun 2023

Dear Dr. Piccolo,

Thank you very much for submitting your manuscript "Many bioinformatics programming tasks can be automated with ChatGPT" for consideration at PLOS Computational Biology.

As with all papers reviewed by the journal, your manuscript was reviewed by members of the editorial board and by several independent reviewers. In light of the reviews (below this email), we would like to invite the resubmission of a significantly-revised version that takes into account the reviewers' comments.

It will be specifically important to address the comments from all three reviewers, including reviewer #1.

We cannot make any decision about publication until we have seen the revised manuscript and your response to the reviewers' comments. Your revised manuscript is also likely to be sent to reviewers for further evaluation.

Sincerely,

@bffo

BF Francis Ouellette

Section Editor

PLOS Computational Biology

Reviewer's Responses to Questions

**Comments to the Authors:**

Reviewer #1: This manuscript argues that chatGPT can and should be used in research settings in bioinformatics, and, to a lesser extent, in teaching coding. The work is based on the experience of the authors in teaching an introductory coding class for bioinformatics, using code generated by querying ChatGPT with problems prepared for that class, and investigating the solutions provided via an automated grader used in educational settings.

The exercises the authors provide seem to be very basic introductory programming tasks for teaching basic coding concepts. String handling, control structures, loops, lists, dictionaries, etc. Some are bioinformatics-adapted and the final tasks seem to be more advanced with use of Pandas, CLI parsing and some visualization. However, those more complex tasks, by the authors own data, are the tasks in which ChatGPT failed. Therefore, I do not see evidence that ChatGPT I believe that the title, “Many bioinformatics programming tasks can be automated with ChatGPT”, is misleading, and should perhaps be changed to “ChatGPT receives a C grade on introductory programming tasks”.

Having read this manuscript, I believe it clearly demonstrates why using ChatGPT as a programming aide for scientists is harmful, and why teaching programming with ChatGPT is probably even more harmful. To do so I will start by quoting the first sentence in the Discussion:

“These findings are remarkable and signal a new era for life scientists. For many basic- to moderate-level programming tasks, researchers no longer need to write code from scratch.”

First, I will address this statement that has to do with the suggested use of ChatGPT in scientific coding. I see this statement as highly problematic. First: the authors have shown that for 25% of the tasks, ChatGPT requires several iterations until it generates a correct answer, if ever. Students are expected to have a higher success rate when submitting their exercise. One may argue that the back-and-forthing the authors did with ChatGPT to improve the code might be equivalent to the debugging process prior to grade submission, but it isn’t.

Second, the authors seem to advocate the use of ChatGPT to simplify simple code writing, pretty much the same way a calculator is used to simplify arithmetic: a tool to avoid basic errors, and to get on with the more important tasks of using the code for analysis rather than investing time in writing and debugging the code. Except that a calculator is always right (2+3=5, always) whereas with ChatGPT the answer is 25% wrong as it might vary between 2+3=4 in one iteration and 2+3=6 in another, with various other plausible but wrong results. Relying on ChatGPT-generated code would be like relying on a calculator that generated only 75% of the answers correctly in the first try. That scientist will not have the capability to debug the code. A coding-competent scientist may be tempted to use ChatGPT to write the code, and gloss over the details of the result. If one sees ChatGPT as a standard of truth, without understanding the code at all, one does not have the necessary skills to debug. Debugging, and not initial code writing, is the bulk of programming time, effort, and ability as coding is mostly about debugging and further adapting code to changing libraries, operating system, language versions and user input and requirements. The latter is especially true for research code which is used for exploratory analysis. If one does not have the skills to debug, one will not have the skills to modify code as required when the analysis requires changing, when the format of the input data changes, or any other myriad issues. They will have no tools for creating software tests that look for things such as edge cases and other problems that may arise with ChatGPT-generated code that is plausible, yet will fail, and the scientist generating the code will not even know that it failed, as the output will be plausible, but wrong. In a good scenario, the code generated by ChatGPT may not be completely wrong, and provide correct analysis to input, but only in some cases.

Finally, I don’t see any of the tasks they provided to ChatGPT as “moderate”. They are all introductory and easy. Most have little real-world relevance, although they have great educational value in teaching basic coding.

As for the educational aspect: here the authors seem to be more reserved. Quote: “Our findings have important implications for education. Until LLMs demonstrate an ability to replace all human programming efforts, it will remain necessary for students (and others) to gain programming skills”. This statement has the implicit assumption that programming skills have a threshold level that needs to be acquired. I disagree with that assumption, as programming is an iterative process that requires domain knowledge, creativity, synthesis, and many other skills. There is no threshold, there is constant skill acquisition in many dimensions.

However, I do agree with the authors that programming skills should be gained and that students should be able to do them themselves without restoring to a code generating LLM. To go back to the calculator analogy, one needs to know how to use a pencil-and-paper addition or long division, even if one will hardly ever use it in the future, otherwise one is arithmetically illiterate. The auction they advocate here is well-placed. It is therefore important to educate programming without the use of an LLM, as the authors suggest.

To go to the Conclusions: “In this study, we provide evidence that dialogue-based LLMs, such as ChatGPT, can aid in solving programming exercises in the life sciences, particularly in the field of bioinformatics.” I disagree. The authors have not demonstrated that they have provided anything but very basic problems to chatGPT, and I am not sure that they are research or real-world-relevant bioinformatics tasks but rather very basic programming exercises.

“However, despite generally excellent performance, these models cannot replace the need for human programming efforts entirely.”

75% is not “excellent”, it is a “C” grade. Especially on the very simple problems that the authors placed in the Additional Materials.

“In an authentic research setting, where an auto-grader is not available to provide instant feedback on the correctness of model-generated code, there is a risk that relying on their outputs may produce erroneous results.” I agree. And it may be harmful.

“With the help of machine learning models, instructors can provide more personalized and efficient feedback to students, and researchers can accelerate their work by automating programming tasks”

There is no demonstration of the former regarding students (how will instructors provide more personalized and efficient feedback to students using an LLM rather than grading software? The LLM in question was used for task solving, not for task feedback). As for the latter, using a code generator that is 25% wrong on even the most simple tasks (which are not even applicable in an authentic research or other real-world setting), does not accelerate research. I suspect that it probably slows it down and introduces unnecessary errors.

My suggestion is for the authors to review the data and their conclusions. It seems to me that the results clearly demonstrate that the use of an LLM for generating basic code, and worse, for teaching programming, should be highly discouraged.

Minor comments:

Please provide the URL for the OBF repository in the Data Availability Section

Please provide Results.html also as an Excel sheet or an ods file. It is hard to read as HTML.

Which Version of ChatGPT (dated, if possible) and Python were used?

Reviewer #2: The manuscript from Piccolo, Denny, Luxton-Reilly, Payne and Ridge explains the significance of large language models like ChatGPT applied to bioinformatics programming tasks. The methods explain how the researchers tested ChatGPT on several tasks, and evaluated its performance and the ability to correct answers guided by human prompts. The researchers found that ChatGPT solved 97.3% of the tasks within 7 or fewer attempts.

I believe the study to be very timely and interesting, but I have some suggestions on how to clarify its purpose:

- I would change the title to be less specific to ChatGPT and more general about LLMs. I understand that ChatGPT is well-known and highly discussed at the moment, and that this is the LLM that was used for the study, however I believe that the results of the study are applicable beyond ChatGPT only, and that keeping it in the title exposes the manuscript to the danger of "early obsolescence". This is just a suggestion and I would leave the decision to the authors preference, also considering their sentence in the discussion about the limitations of the study applicability.

- In the introduction, I believe that the purpose of the study should be better clarified. Was it to test if bioinformaticians' job can be sped up? Do the authors think they could quantify this, comparing the time to formulate ChatGPT prompts and to write code independently?

- Or was the purpose to discuss ChatGPT's potential as a learning tool? The introduction mentions that "learning to

program is a daunting challenge for many researchers". To discuss LLMs' implications in this field, I believe more data should be provided about the level of experience of the humans providing prompts to ChatGPT in this study. Were they all experts, even though they "took the stance of a naive programmer"? If so, would ChatGPT performance result change if the prompts were formulated by real beginner programmers?

- Even though all the conversations between humans and ChatGPT are shared, I would appreciate more summarizing considerations about the prompts to correct ChatGPT's previous errors. I don't find the explanation exhaustive: "our feedback focused on helping ChatGPT understand the exercise requirements and was restricted to natural language such that no source code was present in any of our prompts". Was the type of error (generated by the previous version of the code) provided? If not, and if it was rather an effort to rephrasing the prompt in such a way that it could better understood by the LLM, how much expert knowledge was involved in this process? And how did the researchers decide between the two alternatives: "helping ChatGPT understand the exercise" and "indicated that the previously generated solutions were unsuccessful and asked ChatGPT to try an alternative approach"?

I believe that the additional details suggested above would strengthen the message of the paper. In addition, the ambiguity I find in the introduction (is the purpose to demonstrate ChatGPT efficacy in learning or in working tasks?) reflects in some conflicts I find in the final discussion.

- If the purpose is to demonstrate that bioinformaticians will work quicker with ChatGPT, if we assume that they will all do once they get a job in the field, and if the purpose of the course is to prepare them to their future working environments, why summative assessments at the University do not allow access to the Internet?

- In relation to the above, the manuscript comments on the danger of "over-reliance by novices" and this suggest that this is the reason why summative assessment still don't incorporate the use of LLMs. Hence, the discussion seems to indicate that LLMs are not considered by the authors (yet) a good tool for learning, but this is not what suggested by the introduction (e.g. "learning to program is a daunting challenge for many researchers").

- I would also be interested in knowing more about the suggestions "one way that instructors could counter this behavior is to use LLMs to generate student-specific questions about their code" and "when students attempt to devise solutions but become stuck, they may be able to use LLMs as intelligent tutors".

In summary, I believe the study to be relevant, well structured and deserving to be published. However, I would appreciate a better clarity on its premises, aims, and conclusions from the point of view of the authors. Giving the inherently subjective nature of my comments, I am anyway happy for the authors to incorporate my suggestions to the extent they see fit and I would accept the submission even with minor or no changes.

Reviewer #3: The work tests the use of LLM, chatGPT specifically to generate bioinformatics relevant programming code. The authors conclude that 1)the chatGPT solutions are generic and can fail in providing correct solutions for specific biology problems 2) human in the loop is required to evaluate the solutions 3) since the LLM models are easily accessible the educators will have to account for their usage by themselves and by their students and adapt teaching and assessments accordingly.

It would be good to provide a clear table of DOS and DONTS

The versions of GPT model used must be mentioned

A big gap in the study is to show comparison with another LLM model - it is likely that the responses will vary between models and it is important to highlight those differences due to prompt, model types and model training.

While LLM models can be used in assisting with coding tasks, it does require the a human in the loop to evaluate the outcomes. The current solutions are not self sufficient.

**Have the authors made all data and (if applicable) computational code underlying the findings in their manuscript fully available?**

Reviewer #1: Yes

Reviewer #2: Yes

Reviewer #3: Yes

PLOS authors have the option to publish the peer review history of their article (what does this mean?). If published, this will include your full peer review and any attached files.

Reviewer #1: No

Reviewer #2: **Yes: **Lisanna Paladin

Reviewer #3: No
---

## [Decision Letter · Decision Letter 1]

12 Sep 2023

Dear Dr. Piccolo,

We are pleased to inform you that your manuscript 'Evaluating a large language model’s ability to solve programming exercises from an introductory bioinformatics course' has been provisionally accepted for publication in PLOS Computational Biology.

Best regards,

@bffo

BF Francis Ouellette

Section Editor

PLOS Computational Biology 

Patricia M Palagi

Section Editor

PLOS Computational Biology 

Reviewer's Responses to Questions

**Comments to the Authors:**

Reviewer #1: The authors have carefully read and answered all of my previous remarks. I appreciate their thorough responses, and the sweeping changes they made to the paper. However, we sill have a fundamental disagreement as to the credible use of ChatGPT. I will address what I see as the crux of the matter.

"As described in our earlier comments, we believe that iteratively generating code and

receiving feedback on functional correctness is a valid approach both in an educational

setting and in research settings".

My question is: why invest time and effort in developing the dubious "skill" of ChatGPT prompt engineering, rather than teach the real skill of writing and debugging a program? I do not see prompt engineering as formative learning associated with programming, it is simply a highly idiosyncratic way to get the generative-AI du-jour to perform adequately. While debugging is a large part (perhaps the biggest part) of learning how to program. I'm at a loss as to how anyone can learn how to program, without this fundamental skill. Throwing stochastically generated code lines at a code-checker and tweaking the promptings until converging onto a solution is not, in my opinion, a valid way to teach programming. Especially when in real-life situation there will not be such a code-checker, and the programmer would need to know how to generate a proper test set and test coding by themselves. The authors seem to imply that debugging can be introduced afterwards, in a more advanced teaching setting, but I do not believe that to be correct. Proper debugging skills are fundamental to coding and should be practiced at every level.

"Without approval we cannot provide details"

Although that is understandable, it is unfortunate, as it hampers both the authors' ability to defend their claim, and mine to accept it.

ChatGPT's specious stochastic output can be highly tempting for the beginning (and not-so-beginning) programmer to accept at face value, even though the authors have demonstrated that it is 25% erroneous. It is no different than any other fact-finding use of ChatGPT, which, left unscrutinized (or scrutinized in a cursory or wrong fashion) can lead to disastrous results due to chatgpt's "hallucinatory" behavior.

My question is: why teach programming using a stochastic code generator that is known to be erroneous? Would that not be like teaching arithmetic using a calculator that sometimes outputs 25+32=56 and sometimes 25+32=54? Would it not be better to do without the calculator altogether and teach long addition properly?

"Thank you for this suggestion. We attempted to use Excel, but the number of characters in

some of the programming prompts exceeded the maximum limit for a cell, so the prompts

were truncated. Thus, we used HTML. Although the HTML solution has disadvantages, wefeel it is a reasonable compromise."

A csv or tsv file would probably be easier to parse, then. HTML would be good for visualization of the results, but not for making them FAIR.

Reviewer #2: Thank you for taking into consideration my comments and suggestions.

Reviewer #4: This well structured report describes a study to investigate the potential of LLMs in aiding life scientists with computer programming tasks to help learning. Given that programming is an essential tool for life scientists but remains a challenging skill to master, this study evaluated how well ChatGPT can interpret and generate functional code based on human-language prompts in a training exercise setting.

A key issue is that naïve-to-coding students will inevitably use LLMs as they learn to code – are they of any help? Will the LLMs be able to provide appropriate code?

The results of this study details the responses and potential for LLMs in assisting with programming skills.

The manner in which teaching of coding will be impacted relies upon better understanding how well LLMs can code from prompts. Here the work makes a contribution in the setting of bioinformatics-related coding exercises

The majority of the other reviewers concerns have been addressed, and the concerns I may have raised have been reflected in their comments and responses.

**Have the authors made all data and (if applicable) computational code underlying the findings in their manuscript fully available?**

Reviewer #1: Yes

Reviewer #2: Yes

Reviewer #4: Yes

PLOS authors have the option to publish the peer review history of their article (what does this mean?). If published, this will include your full peer review and any attached files.

Reviewer #1: No

Reviewer #2: **Yes: **Lisanna Paladin

Reviewer #4: No

---

## [Editor Report · Acceptance letter]

25 Sep 2023

PCOMPBIOL-D-23-00520R1 

Evaluating a large language model’s ability to solve programming exercises from an introductory bioinformatics course

Dear Dr Piccolo,

I am pleased to inform you that your manuscript has been formally accepted for publication in PLOS Computational Biology. Your manuscript is now with our production department and you will be notified of the publication date in due course.

With kind regards,

Judit Kozma
